# Chronic Hepatitis C Cascade of Care in Prisoners—Is There Still Some Work to Do? Analysis of Two Large Penitentiaries in Northern Italy

**DOI:** 10.3390/ijerph21010104

**Published:** 2024-01-17

**Authors:** Anna Cambianica, Valentina Marchese, Francesca Pennati, Alessandro Faustinelli, Manuela Migliorati, Fabio Roda, Angiola Spinetti, Serena Zaltron, Simona Fiorentini, Arnaldo Caruso, Eugenia Quiros-Roldan, Francesco Castelli, Emanuele Focà

**Affiliations:** 1Department of Clinical and Experimental Sciences, Section of Infectious and Tropical Diseases, University of Brescia, 25123 Brescia, Italy; f.pennati@unibs.it (F.P.); a.faustinelli001@studenti.unibs.it (A.F.); m.migliorati022@unibs.it (M.M.); eugeniaquiros@yahoo.it (E.Q.-R.); francesco.castelli@unibs.it (F.C.); emanuele.foca@unibs.it (E.F.); 2Division of Infectious and Tropical Diseases, ASST Spedali Civili Hospital, 25123 Brescia, Italy; valentina.marchese@ymail.com (V.M.); angiola.spinetti@hotmail.it (A.S.); zaltron.serena@tiscali.it (S.Z.); 3Unit of Prison Health, ASST Spedali Civili Hospital, 25123 Brescia, Italy; fabio.roda@asst-spedalicivili.it; 4Unit of Microbiology and Virology, Department of Molecular and Translational Medicine, University of Brescia, 25123 Brescia, Italy; simona.fiorentini@unibs.it (S.F.); arnaldo.caruso@unibs.it (A.C.)

**Keywords:** hepatitis C, HCV, direct-acting antivirals, DAA, prison, inmates, screening

## Abstract

Penitentiaries have a higher burden of communicable diseases compared to the general population. Prisoners should be tested for hepatitis C virus (HCV) and have direct access to treatment. We analysed the HCV cascade of care in two penitentiaries in Brescia, Northern Italy. At admission, prisoners are offered a voluntary screening for HCV, while patients with known infections are tested with an HCVRNA measurement. We performed an observational retrospective study including all the subjects admitted to the penitentiaries from 1 January 2015 to 31 October 2021. We conducted a descriptive analysis. During the study period, 5378 admissions were registered, and 2932 (54.5%) screenings were performed. Hepatitis C virus antibody positivity was found in 269 tests (9.2%). Hepatitis C virus RNA was detectable in 169 people. During the study period, 77 treatments with direct-acting antivirals (DAAs) were administered. Follow-up was available in 45 patients, and sustained virological response (SVR) was documented in 44 of them. Retention in care occurred in less than half of the prisoners after release. Our data demonstrate poor screening adherence that could benefit from educational programs. Treatment rates could be improved with test-and-treat programs. More efforts are needed to eliminate HCV as a public threat by 2030. Dedicated local networks, including infectious diseases (ID) departments, substance abuse services and prisons, could mitigate these issues.

## 1. Introduction

Penitentiaries are known for a higher burden of communicable diseases, including hepatitis C virus (HCV), hepatitis B virus (HBV), human immunodeficiency virus (HIV) and tuberculosis, compared to the general population. This is strongly associated with the high occurrence of injecting drug habits and high-risk sexual behaviours in the prison population, together with penitentiary overcrowding and poor sanitary conditions [1,2,3,4].

According to international guidelines, prison inmates should be routinely tested for HCV and have rapid access to treatment, as well as be provided with opioid substitution therapy when necessary [1,5]. Thanks to the introduction of direct-acting antivirals (DAAs), eradicating HCV during incarceration is now easier and requires shorter treatment courses than in the interferon (IFN) era. Nonetheless, HCV treatment can be difficult to administer due to frequent transfers to other services, short prison stays and sometimes patient rejection [6,7].

Moreover, ensuring the continuum of care is a challenge: the majority of patients are often lost to follow-up, especially after release, when the persistence of dangerous behaviours predisposes to reinfections [8].

However, up-to-date and precise data regarding HCV epidemiology in the prison setting are lacking, as well as information about treatment administration and eradication of the infection: available estimates are neither nationwide nor inclusive of the detainee population. Because of that, it is difficult to estimate the size of the problem and therefore implement diagnostic and treatment programs [9,10,11]. 

In Italy, screening and treatment for chronic hepatitis and HIV are provided free of charge and jointly suggested by the Ministry of Health and Ministry of Justice, although differently implemented based on local health system organizations [12,13]. Nevertheless, data are scarce for epidemiology, the efficiency of screening and treatment enrolment: most of the epidemiological studies were performed before the DAAs era and do not represent the current epidemiological situation. Due to the lack of national data, few local epidemiological data are available [14,15,16].

Therefore, there is a need to evaluate the HCV cascade of care in the penitentiary environment to assess the rate of HCV screening acceptance and the burden of HCV-Ab positivity. At the same time, it is important to explore the DAAs treatment offer and the proportion of HCV eradication. 

Our aim was to describe the HCV cascade of care in two penitentiaries in Brescia Province, Northern Italy. We assessed the seroprevalence of HCV-Ab among inmates and the proportion of HCV-related liver disease. Moreover, we evaluated the antiviral treatment proposal as well as the rate of early and sustained virological response (SVR) after treatment. Lastly, we described the retention in care rate after release.

## 2. Materials and Methods

### 2.1. Study Setting and Organization of Health Assistance

Brescia, Northern Italy, is the largest Italian province, accounting for 1.266.000 million of Inhabitants (ISTAT 2020). The city of Brescia has one jail (Casa Circondariale) that welcomes people awaiting trial or convicted with short-term penalties (inferior to five years), and one prison (Casa di Reclusione) where women and those who committed major crimes and have longer penalties are confined. Together, they have a cumulative capacity of 260 prisoners, although overcrowding is a chronic issue. 

At admission, a voluntary serological screening for HCV, HBV, HIV and syphilis is offered to all prisoners. Hepatitis C virus RNA instead of HCV-Ab is measured when a previous diagnosis of HCV infection was made. 

In line with national indication, multidisciplinary health assistance is granted within the penitentiary structures through an agreement with the local tertiary public hospital (Azienda Socio-Sanitaria Territoriale (ASST) Spedali Civili, General Hospital of Brescia) [13]. Infectious disease specialists perform in-prison evaluations of those people with positive screening results or known infection at least once every two weeks, with the possibility of increasing the number of consultations on demand. 

Serologies, viremia and blood chemistry tests are carried out at the in-hospital laboratories, while the ultrasound examination for disease staging is performed within the prison facility. Other instrumental examinations (i.e., computed tomography, magnetic resonance imaging liver elastography, and esophagogastroduodenoscopy) are performed at the hospital.

Prescription of DAA treatments is authorized through the Italian Medicines Agency portal (Agenzia Italiana del Farmaco, AIFA) and distributed at the penitentiary health facilities, while other possible treatments (i.e., ligature of oesophageal varices, locoregional treatment of hepatocarcinoma or surgery) are performed at the hospital. 

Continuum of care in Brescia city is granted thanks to electronic medical records (NetCare—Healthware^TM^). Patient files are linked to that of the ID Unit, ensuring record availability after release. Patients residing in the Brescia province are referred to the infectious diseases (ID) outpatient clinic in ASST Spedali Civili General Hospital, Brescia; otherwise, access to the nearest ID Unit is recommended at release. 

Direct-acting antivirals have been offered since 2015, in line with the national guidelines, while pegylated interferon (Peg IFN) was prescribed before. Treatment options include sofosbuvir/velpatsavir, glecaprevir/pibrentasvir, grazoprevir/elbasvir and others, prescribed according to national indications and international guidelines [6]. A new criterion for the prescription of DAAs was released by AIFA in October 2019, issuing that, in settings with limited access to healthcare services (as penitentiaries), liver fibrosis scores as aspartate aminotransferase (AST) platelet ratio index (APRI) or FIB-4 can be used instead of liver elastography to assess liver fibrosis and to prescribe treatment accordingly [17,18].

### 2.2. Study Methodology and Ethical Clearance

We performed an observational retrospective study including all the subjects admitted to the two penitentiaries from 1 January 2015 to 31 October 2021 with a diagnosis of HCV, either achieved through serological screening or already known. Microbiological and virological data were retrieved from prison records and from the Microbiology and Virology Unit of the ASST Spedali Civili General Hospital, while sociodemographic data and information about therapies were retrieved from the ID medical records. All data were collected in an Excel file. 

We performed a descriptive analysis. Categorical variables were expressed as percentages, numerical variables as means or medians and interquartile range (IQR), as appropriate.

The Ethics Committee of Brescia Provence approved this study in accordance with Legislative Decree no. 211 of 24 June 2003, as well as subsequent additions and authorizations. We conducted the research with full respect for human dignity and fundamental rights as dictated by the “Declaration of Helsinki” and by the standards of “Good Clinical Practice” (GCP) issued by the European Community (as implemented by the Italian Government and in accordance with the Guidelines issued by the same bodies). We implemented what the Council of Europe Convention for the Protection of Human Rights and Dignity of the Human Beings provided in the application of biology and medicine (Oviedo on 4 April 1997). 

We used alpha-numeric code to anonymize patients’ data in observance of the rights provided for by privacy legislation (Legislative Decree no. 196/2003 Art. 7). Informed consent has not been provided since this study was retrospective and non-pharmacological, and in Italy, ethical clearance for these studies is not needed (Italian Guidelines for classification and conduction of observational studies, established by the Italian Drug Agency, “Agenzia Italiana del Farmaco—AIFA” on 20 March 2008). In addition, we used the general authorization of the Italian Guarantor for the use of anonymized demographical and clinical data.

This study has been approved by the Ethics Committee of the Province of Brescia (number of protocol: 3981).

### 2.3. Laboratory Methods and Clinical Assessment

Serological tests performed over a six-month interval in the same person were considered a new screening due to possible re-admission.

Hepatitis C virus RNA was measured using polymerase chain reaction (PCR) and was defined as detectable when the patient had more than 15 International Units (IU) per millimetre of blood. 

The following blood tests were performed to assess the stage of disease: alanine amino transferase (ALT) and AST, total and fractioned bilirubin, coagulation time, full blood cell count, and abdominal ultrasound. Hepatic fibrosis was either deduced from liver stiffness, measured with hepatic elastography, or assessed with APRI score ((AST value/upper limit of normal AST range)/platelet count x 100)_. A liver stiffness > 12.5 kPa or an APRI score > 1.0 were used to define cirrhosis.

Sustained virological response was defined as HCVRNA undetectability 12 or 24 weeks after the end of treatment. Hepatitis C virus recurrence (both reinfection and late relapse) was defined in the case of the reappearance of HCVRNA in treated patients after achieving sustained virological response (SVR). Reinfection was suspected in cases of a recurrence of HCV infection occurring more than 12 or 24 weeks post-SVR, in the case of continuous at-risk behaviours, or diagnosed, in the case of different genotypes.

## 3. Results

### 3.1. Admission and Screening 

During the study period, 5378 accesses were registered in the two penitentiaries. At admission, 2932 (54.5%) HCV-Ab tests were performed, with a mean of 418.9 tests per year. We observed a slight decrease during 2020 when only 41.3% (317/767) of accesses were screened. HCV-Ab positivity was found in 269 tests, which represented 9.2% of screenings. Table 1 describes the number of people who were admitted to prison, screened and tested positive for HCV antibodies divided by year.

After checking for multiple screenings due to readmission, HCV-Ab positivity was identified in 189 patients. Previous serological status was unknown in 51/189 (27.0%) people, while, among those with a previous screening available, 107/189 (56.6%) were aware of HCV infection; a new diagnosis was made in 31/189 (16.4%).

In all inmates with positive serology finding at screening, HCVRNA was measured in 79.9% (151/189) and was detectable in 62.9% (95/151). Taking into consideration new HCV diagnosis only, HCVRNA was positive in 26/31 (83.9%) cases. 

Overall, HCVRNA was measured in 263 inmates (including both those with a previously known positive serology and newly diagnosed) and was detectable 169/263 (64.3%); in 56.2% of them (95/169), HCVRNA measurement was performed after HCV-Ab screening, while 43.8% (74/169) were already aware of chronic HCV infection. Genotype analysis was performed in 151/169 (89.3%) patients and showed a high prevalence of genotype 1a (47.0%) and 3a (27.8%), followed by 4a/4c/4d (10.6%). Figure 1 shows genotype distribution.

### 3.2. Patients Characteristics

Of 263 inmates tested for HCVRNA, 243 of them were males (92.4%). The great majority of people were born in Italy (215/263, 81.7%), while the others were born in the Middle East (13/263, 4.9%), Africa (12/263, 4.6%), East Europe (10/263, 3.8%), South America (4/263, 1.5%) and Western Europe (3/263, 1.1%). Nationality was unknown in six cases (2.3%). The median age at admission was 45 years and varied from 20 to 79 years (IQR 14.5 years). Risk factors for HCV infection were unknown in 156/263 (59.3%) patients, the others reported intravenous drug use (99/263, 37.6%) and/or unprotected sexual intercourse (10/263, 3.8%). Other risk factors, such as tattoos, blood transfusion or vertical transmission, were seldom reported. Human immunodeficiency virus coinfection was found in 48/263 (18.3%) patients, 72.9% (35/48) of which had detectable HCVRNA. On the contrary, HCV infection has been cleared or previously eradicated in 13/48 (27.1%) HIV-infected patients. 

Information about HBV coinfection was available in 136/263 (51.7%) patients, 3 (2.2%) of which had detectable HBV surface antigen (HBsAg). 

Of 263 patients, 2 (0.8%) developed hepatocellular carcinoma (HCC) during the study period. Both of them reported persistent intravenous drug use and alcohol abuse. One patient had a long history of cirrhosis and developed HCC six months after the end of DAA treatment. The second one had been previously diagnosed with HIV and chronic HBV infection, but he had always refused treatment and medical care. At prison admission, he was diagnosed with HCV and advanced liver disease complicated by HCC; he therefore could not receive antiviral therapy. Both of them received locoregional treatment at first, but multifocal HCC relapsed months after. Chemotherapy was prescribed, but the two patients stopped it because of adverse drug events. The first patient is currently lost to follow-up, and the second one died three years after the HCC diagnosis.

Assessment for hepatic fibrosis using elastography was available in 37.6% (99/263) of the patients. Median stiffness at hepatic elastography was 6.5 kPa (IQR 6.1 kPa), while the median APRI score was 0.625 (IQR 0.8), both indicating mild fibrosis. 

### 3.3. Treatment

Previous treatment history at admission was available in 165/263 (62.7%) people: 41 (15.6%) had received PegIFN plus ribavirine, 7 (2.7%) had been administered both PegIFN and DAAs, 2 (0.8%) received only DAAs. The former therapeutic regimen was unknown in 2 (0.8%) patients, while 113 (43.0%) patients were naïve to antiviral treatment. Information about previous therapies was not available in 99 (37.6%) patients. 

During the study period, 77 treatments with DAAs were administered out of 169 active infections, corresponding to 45.6% of people affected by chronic C hepatitis with access to a cure. 

The median time between the first HCVRNA measurement and treatment administration was 13 months, with a minimum of eight days to a maximum of 91 months (IQR 30 months). This value gradually decreased from 2015 (median time: 39.9 months) to 2021 (median time: 2.8 months).

Sofosbuvir/velpatasvir was administered in 41/77 (53.2%) cases, and on one occasion, it was associated with ribavirine; glecaprevir/pibrentasvir was given in 27/77 (35.1%) cases. Other combinations, such as grazoprevir/elbasvir or sofosbuvir/daclatasvir/ribavirine, were administered in a minority of cases (Figure 2). A twelve-week course of treatment was preferred to eight-week and twenty-four-week ones, administered in 49/77 (63.6%), 22/77 (28.6%) and 6/77 (7.8%) patients, respectively. Early treatment interruption was observed on two (2.6%) occasions, one of which needed re-treatment. The number of treatments per year increased from three in 2015 to 16 in 2018 and remained steady afterwards, with a minor deflection in 2020, when 11 therapies were administered. 

Sustained virological response at 12 and/or 24 weeks was documented in 44/45 treatments for which follow-up was available (97.8%), but it was not possible to assess it in 32/77 (41.6%) administered treatments. Figure 3 shows the number of patients who were eligible for treatment, received treatment and eradicated HCV infection. 

Information on SVR was not available for the following reasons: three people were transferred to other institutions before the end of follow-up, while two inmates ended treatment recently and are currently on follow-up. In 8/32 cases, HCVRNA was undetectable at week four after the end of treatment, while in 5/32, HCVRNA was undetectable at the end of treatment; none of them was monitored afterward. Figure 4 summarizes the cascade of care for HCV infection.

Reinfection was described in three patients. Of the three reinfected patients, it should be noted that three (100%) reported active intravenous drug abuse and two (66.7%) had HIV/HCV coinfection. Reinfections were diagnosed during follow-up as outpatients at our clinic. The mean time between the end of treatment and reinfection diagnosis was 28 months. At the end of the study, they were still awaiting re-treatment.

### 3.4. Retention in Care

Information about retention in care after release was available in 90 cases: it occurred in 40 patients (44.4%). Of the 50 patients lost to follow-up, 25 (50.0%) had received treatment during detention or had negative HCVRNA at admission, while 25 (50.0%) had detectable HCVRNA at release and, therefore, still needed treatment. Among the patients lost to follow-up, six (12.0%) had advanced liver disease.

## 4. Discussion

In this study, we observed poor adherence to HCV screening at admission, with only half the inmates being tested. Moreover, we highlighted a slight decrease in the number of screenings during 2020. The retrospective nature of the study prevented us from assessing the reasons behind the lack of adherence to screening, although in the first year of the COVID-19 pandemic, the temporary interruption of consultations and the focus on screening procedures for COVID-19 could have been responsible for the decrease.

Hepatitis C virus screening should be implemented to reach the WHO goal of eradicating HCV infection by 2030, as it represents the only strategy to diagnose asymptomatic patients, and it is recommended by the National Italian Institute of Health [19]. 

To accomplish this, more information about the reasons behind poor adherence to screening should be obtained. People refusing screening could be administered a questionnaire investigating their motives to allow the identification of specific determinants of refusal and, therefore, the ability to properly address them.

As other experiences showed, provider-initiated strategies for blood-borne virus screening, such as the one implemented in our setting, have proven more effective than client-initiated ones [1]. Moreover, peer-to-peer communication and education programs, together with the use of rapid salivary and capillary tests, were able to greatly increase HCV screening acceptance [14,20]. Nevertheless, the serological screening we adopted has the advantage of allowing testing for other infections that share the same routes of transmission. Rapid tests could be used as a preliminary tool to screen for HCV-Ab. Hepatitis C virus RNA measurement and genotype analysis could be performed later, together with other screening using a traditional blood test, therefore speeding up the diagnostic process. The choice to proceed with non-invasive tests could benefit from targeted qualitative studies in the inmate population to assess the acceptability of screening tools. This, combined with information programs, could contribute to increasing screening adherence. 

On top of that, HCV screening could be performed every six months during detention, allowing people who previously refused it to be tested. Moreover, it could be used as a tool to diagnose the infections acquired during imprisonment. 

In this study, we found that HCV seroprevalence in a prison setting is higher than in the general population in Italy and in Western Europe (9.2% vs. 2.0% and 0.9%, respectively) [21], although we observed lower prevalence rates than other studies (9.2% vs. 9–90%) [16,22]. Our seroprevalence may, however, be underestimated since we considered only positive screenings and not people with a known infection (who, therefore, did not undergo HCV-Ab screening at admission). Notably, 43.8% of active diseases were detected among inmates having a known status of infection, including some with a positive screening available in previous imprisonments in the same facilities.

As highlighted by other experiences, we confirmed a close relationship between injecting drug habit and HCV infection, with more than a third of patients reporting substance abuse as a risk factor [4,22]. Genotype analysis showed a prevalence of genotype 1a and 3a, which are commonly associated with intravenous drug use. On top of that, HIV and HCV coinfection was present in 18.3% of inmates. The association between HIV and HCV is well known as they share the same routes of transmission [23]. 

Two patients who had high-grade liver fibrosis developed HCC during the study period; such a low incidence could be due to median low-grade fibrosis, as the stiffness values and APRI score showed. Recent literature has confirmed that the prison population is younger and, therefore, has a shorter course of disease compared to the general population, leading to a lower prevalence of advanced liver disease [16]. This underlines the primary importance of early treatment to prevent disease evolution and complications [24].

Worldwide treatment rates among prisoners are low for several reasons: high mobility (transfer, short detections) of inmates, lack of human resources and specialist consultations in facilities, treatment costs or coordination with local health services [25,26]. During the years, we observed a treatment rate of 45.6%. Even though these data are lower than other experiences, with a considerable range of time between diagnosis and treatment, an increase in DAAs administration per year and a gradual decrease in administration latency could be observed at the end of the study [20,27,28,29]. Two factors probably influenced these observations over the years: the availability of additional infectious diseases specialists to perform in-prison consultations, and the changes in the National indication for prescription and offer of free-of-charge treatment. Universal access to treatment for all HCV chronic patients in Italy was granted in 2017; it was previously accessible only for patients with advanced disease since 2015. However, treatment prescription became easier after 2019, when the APRI score was accepted as an indirect measure of liver fibrosis instead of hepatic elastography in settings with limited access to healthcare services. This increased access to treatment, as observed in other experiments described in the same period [29].

Despite that, in the Italian context, genotyping is still a prerequisite for the prescription of eradicating treatment. Genotype analysis, despite being useful in a population with such a high risk of reinfection, is partially responsible for treatment latency. According to recent guidelines, simplified treatment prescriptions without genotype determination can be made when it limits access to care [6]. An alternative to omitting genotype analysis could be performing it at the start of treatment, at least in settings with mobile populations or limited access to healthcare services, as already recommended for liver fibrosis assessment. Finally, prisoners’ high mobility is often a deterrent to starting DAAs. An increased cooperation between detention facilities and medical personnel could improve access to treatment for patients who are transferred immediately after the screening. 

More than half of the patients were lost to follow-up, many of them while still awaiting treatment. Retention in care and therefore treatment administration could benefit from a fast-track strategy with point-of-care testing at our department immediately followed by a medical visit for people who have recently been released.

Sustained virological response could be demonstrated in half of the treated patients, given the high rate of loss to follow-up. Nevertheless, among those for which it was assessable, it was similar to that of other experiences [27,30]. The most recent recommendation to assess eradication at 12 weeks after treatment, rather than 24 weeks, will increase the probability of assessing SVR in such a mobile population. Assessment is crucial from a clinical perspective, especially in inmates and other populations at risk of re-infection. More efforts should be made to guarantee follow-up through a better connection with local services after release. Nonetheless, treatment should not be postponed merely because of the inability to verify SVR, as response rates are notoriously high. Even in the absence of SVR confirmation, a high public health benefit is conceivable. As shown by a previous study conducted in the same area, collaboration with the territorial services involved in addiction care and rehab could be the key to increasing the retention of care [31]. A similar project is currently ongoing in the same area.

The study has some limitations related to the retrospective nature of the design and the collection of real-life data, resulting in some missing data. For the same reasons, it was not possible to assess the reasons behind the refusal of screening, which would certainly have contributed to a better understanding of the obstacles in the screening and treatment cascade.

## 5. Conclusions

Hepatitis C virus seroprevalence in prison is higher than in the general population. Genotype detection and HIV coinfection rates reflect the high injecting drug use habit recorded in the Italian penitentiary system. This highlights the need for preventive and corrective actions, paired with harm-reduction strategies. Screening uptake should be improved, additional studies are required to investigate the reasons behind the rejection.

Treatment administration could benefit from test-and-treat programs. In our real-life experience, simplification of treatment prescription has been demonstrated to be effective in implementing treatment delivery.

Loss to follow-up remains a challenge and needs to be addressed in order to achieve the ambitious goal of eliminating hepatitis as a public threat by 2030. Simplified follow-up procedures (i.e., reduced number of consultations, combined ultrasound, and clinical evaluation) and dedicated local networks, including ID departments, substance abuse services and prisons, could mitigate this issue, both in prison settings and in the delicate phase after imprisonment.

## Figures and Tables

**Figure 1 ijerph-21-00104-f001:**
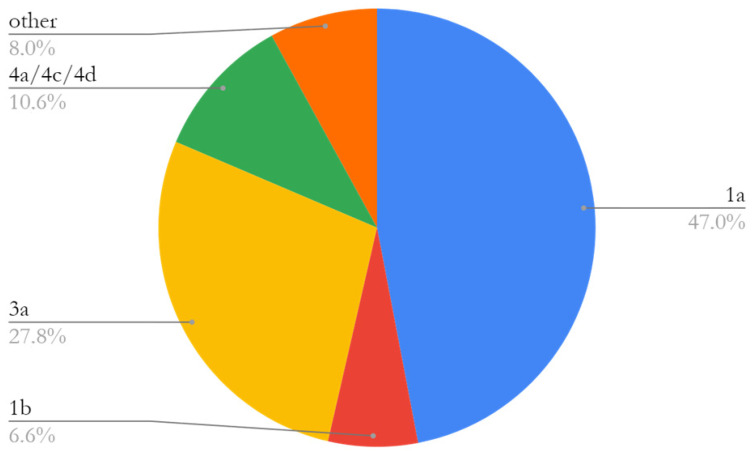
Genotype distribution in 169 hepatitis C virus RNA positive patients.

**Figure 2 ijerph-21-00104-f002:**
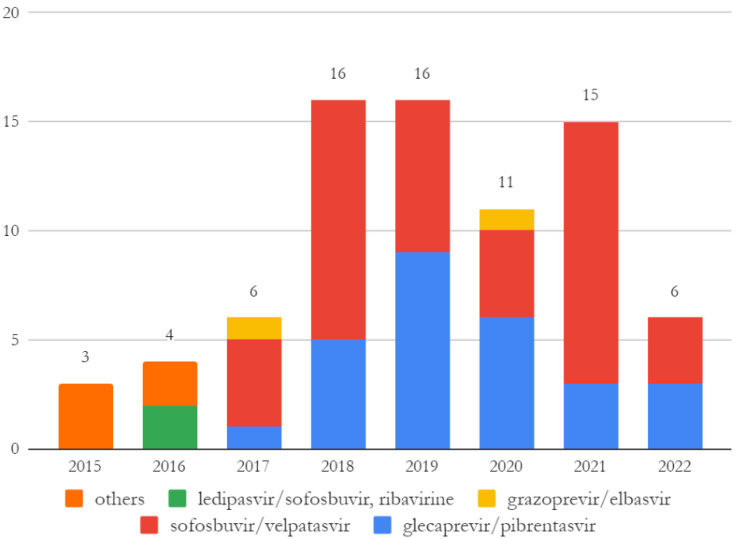
Number of treatments and combination of drugs administered per year. Treatments in 2022 were prescribed in response to screenings performed during the study period.

**Figure 3 ijerph-21-00104-f003:**
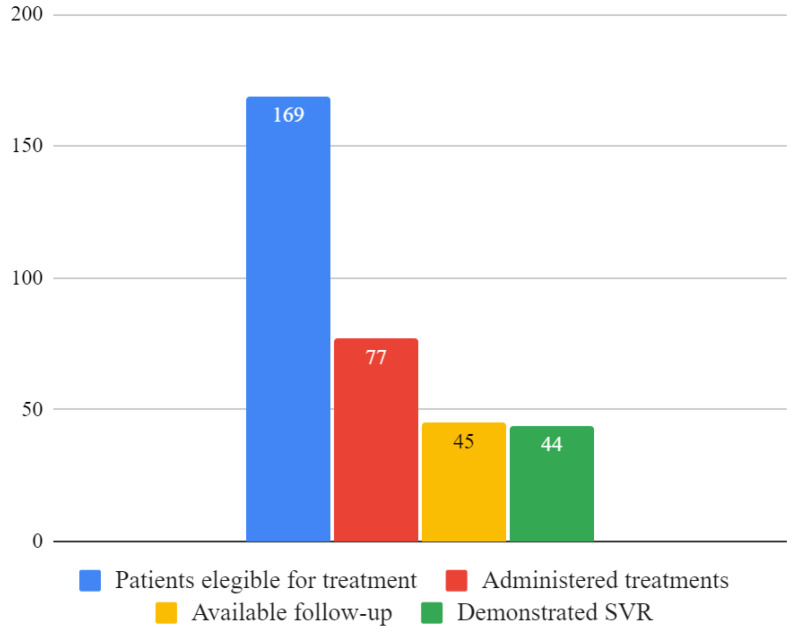
Number of patients eligible for the treatment, administered treatments, available follow-up and sustained virological responses.

**Figure 4 ijerph-21-00104-f004:**
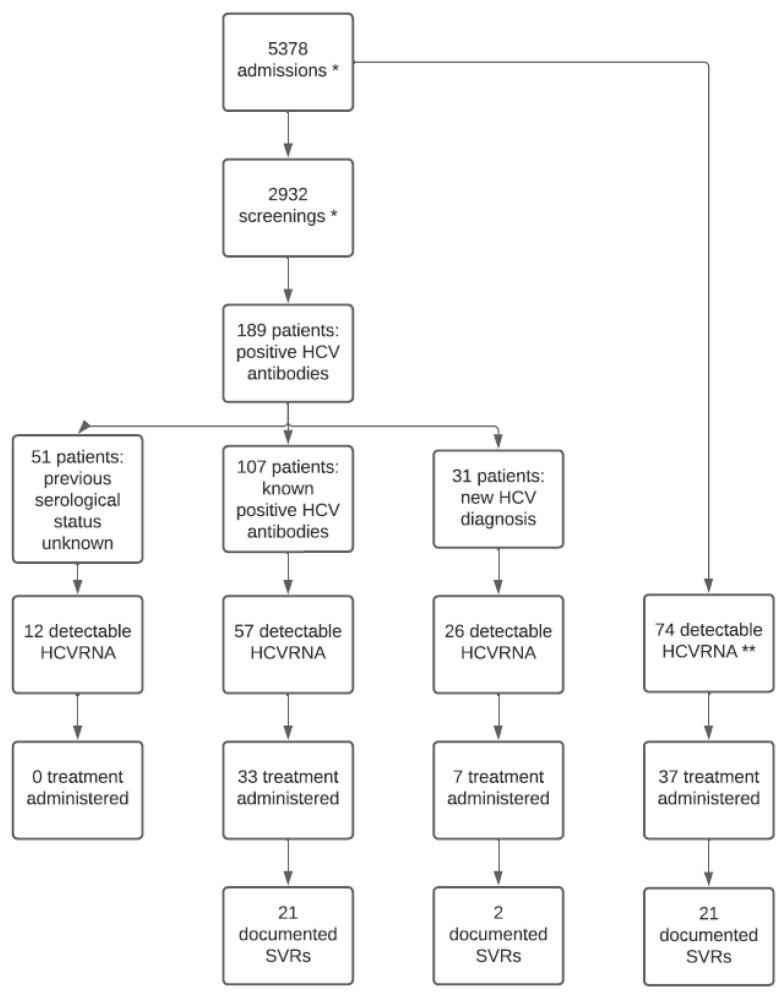
The cascade of care of HCV infection from diagnosis to treatment. * These numbers refer to admissions and screening, some patients may be included more than once. ** Patients already aware of HCV infection at admission and thus directly tested for HCVRNA.

**Table 1 ijerph-21-00104-t001:** Number of admissions to prison, screenings performed at admission and positive serologies per year.

Year	Admissions	HCV Screenings (%)	HCV Positive Antibodies (%)
2015	715	429 (60.0%)	29 (6.8%)
2016	764	457 (59.8%)	43 (9.4%)
2017	767	463 (60.4%)	37 (8.0%)
2018	893	492 (55.1%)	45 (9.1%)
2019	879	462 (52.6%)	41 (8.9%)
2020	767	317 (41.3%)	43 (13.6%)
2021	593	312 (52.6%)	31 (9.9%)
Total	5378	2932 (54.5%)	269 (9.2%)

## Data Availability

The data underlying this article will be shared on reasonable request to the corresponding author.

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
