# Peer review of "Chronic Hepatitis C Cascade of Care in Prisoners—Is There Still Some Work to Do? Analysis of Two Large Penitentiaries in Northern Italy"

_ijerph, 2024, doi:10.3390/ijerph21010104_

Round 1

Reviewer 1 Report

Comments and Suggestions for Authors

Journal: International Journal of Environmental Research and Public Health

Manuscript ID: ijerph-2750834

Title: Chronic Hepatitis C cascade of care in prisoners. Is there still some 

work to do? Analysis of two large penitentiaries in Northern Italy.

Authors: Anna Cambianica et al.

This was one of the important approaches to achieve global HCV eradication. To improve the rate of HCV test and DAA administration, what approaches are needed? Were there any differences in backgrounds between inmates who accepted with HCV test and treatment and those who rejected them? 

Three patients were reinfected. More information should be shown, if possible. For examples, when and where reinfection was diagnosed after the first anti-HCV therapy? 

Author Response

We thank you for the suggestion. We enforced the discussion regarding the approaches to improve HCV screening rate and DAAs administration. We believe that the changes in the National indication for prescription (universal access to treatment for all HCV chronic infected patients and recognition of APRI score as indirect measure of liver fibrosis) have played a pivotal role in increasing treatment administration.

Thank you for highlighting the point concerning the differences between people accepting and rejecting screening. Unfortunately, no data were available regarding people who refused screening. We will certainly keep that in mind for further studies. 

As long as reinfections are concerned, we provided more information along the paper. Two people have recently received re-treatment and obtained SVR (these treatments have not been reported in the paper since they have been administered after the study conclusion), one has not been treated since he was lost to follow-up soon after reinfection. 

Reviewer 2 Report

Comments and Suggestions for Authors

Thank you for the opportunity to read this manuscript detailing the experiences and outcomes of testing people in prison in Italy for HCV.

The work is an interesting read and contributes to the global understanding of how care can be improved in this population.

I have the following questions / comments;

1. Suggest using person first language rather than prisoner or inmate, so eg Person/People in Prison (PIP) or Person/People who are Incarcerated (PWI). This paper gives some rationale for this change in language ‘Mind your language’: What people in prison think about the language used to describe them - Bidwell - 2023 - The Howard Journal of Crime and Justice - Wiley Online Library   (I am not an author of, or have had anything to do with this paper). 

2. Whilst the test uptake reported is not high, it is considerably higher and consistent over a number of years, which is better than many prisons in the UK. It would be interesting to see further work from Italy that explores the reasons behind the low uptake. 

3. It would be interesting to see the test uptake differences between men and women, and between the jail and prison, to see the test acceptability between the different populations and contexts (this difference is emerging in England and it would be helpful to see if the same phenomenon is observed elsewhere as it may identify where different approaches to increasing test uptake could be directed).

4. Interesting also to see a much higher number of people with HCV/HIV co-infection than in the UK. Do you have data on the overall HIV prevalence in the populations, and male/female differences?

5. Were the two people with HCC infected with HCV - had they been treated and if so when? Were there any other factors that increased the likelihood of HCC development?

Thank you.

Author Response

1.We would like to thank you for highlighting this point. We found the paper you attached interesting but decided against changing the terminology as we felt that the changes would have made the article less clear and the reading more difficult. However, if the reviewer feels these changing as necessary, we shall modify the paper accordingly.

2.We totally agree with your suggestion, we will certainly evaluate the possibility of investigating the matter in the next future.

3.We thank you for your valuable observation. Unluckily, the retrospective nature of the study prevented us from assessing these differences, but we will surely keep that in mind for further studies.  

4.We find your observation extremely interesting. The estimated HIV incidence among adults in Italy in 2022 was 3.2 new cases for 100.000 residents, as reported by the National Institute of Health, Rome. It is lower compared to that reported in Western European countries and the European Union (5.1 new diagnoses per 100,000); infection was 3.5 times more common in males than in females [https://www.epicentro.iss.it/aids/epidemiologia-italia].

On the contrary, HIV prevalence in the penitentiary populations seems to be higher, being around 5.1%; almost all the HIV infected people are males [R. Monarca et al., “HIV treatment and care among Italian inmates: A one-month point survey,” BMC Infect. Dis., vol. 15, no. 1, pp. 1–8, Dec. 2015, doi: 10.1186/s12879-015-1301-5.]

5.We thank you for pointing this out, we wrote the required information in the paper (section 3.2 “Patients characteristics”). 

Reviewer 3 Report

Comments and Suggestions for Authors

This is an important paper, and access to data on HCV care in prisons is hard to come by. Overall it is straight forward and well done, but the low rate of HCV screening and some of the other results cast doubt on whether HCV can truly be eradicated in a timely fashion even in a country with essentially universal access to care. I would appreciate a speculation on the part of the authors on why only ~50% of people who are being incarcerated agree to be screened, as well as the slight dip from almost 60% to closer to 40% of the study period. This result really calls for a follow-up questionnaire of people being incarcerated asking for the reason to refuse screening, and whether any educational intervention or 2nd chance at screening 6 months in could improve the overall rate.

Author Response

Thank you for your observations. 

Certainly, HCV screening and DAA administration should be implemented in populations at risk such as incarcerated people and people who inject drugs. 

The reasons behind screening refusal are unknown, although we believe that the fear of being stigmatised after the diagnosis may act as a deterrent. Moreover, many people are unaware of the risk behaviours for HCV transmission and are therefore clueless about being at risk. Lastly, some people may refuse the screening as they are asymptomatic and consider themselves healthy.

We believe that the decrease in the screening rate could be related to the higher number of admissions in 2019 and 2020. This worsened the problem related to overcrowding and hindered the capillary administration of screening. Moreover, screening procedures for COVID-19 and sanitary system reorganization during the first year of pandemic may have put HCV screening in the background.

We found the observation about the questionnaire extremely interesting and provided it in the discussion as a suggestion to implement screening rate. We totally agree with the necessity to repeat screening six months after admission, also because people have high risk behaviours both in prison and outside.